# "I can't make it safe, so I don't do it": Exploring obstetricians' views on barriers and enablers to promoting vaginal birth after caesarean section in Bangladesh

**Aimee Hairon**[1,2]*, **Abdul Halim**[3], **Abu Sayeed Md. Abdullah**[3], **Sumaiya Afroze Khan Atina**[3], **Terry Kana**[4]

**1** Department of International Public Health, Liverpool School of Tropical Medicine, Liverpool, United Kingdom, **2** School of Medicine, University of Liverpool, Liverpool, United Kingdom, **3** Reproductive and Child Health Department, Centre for Injury Prevention and Research Bangladesh (CIPRB), Dhaka, Bangladesh, **4** Faculty of Education, Liverpool School of Tropical Medicine, Liverpool, United Kingdom

* aimeehairon@gmail.com

**Data Availability Statement:** Excerpts of the transcripts relevant to the study are presented in

## Abstract

The rise in the number of facility-based births in Bangladesh has been accompanied by a caesarean section (CS) epidemic. The current CS rate is 45% and while many are performed when medically unnecessary, there is still maternal mortality due to lack of access to CS. A significant contributor to the rising CS rates in Bangladesh is repeat CS. Evidence from high-income settings has shown that vaginal birth after caesarean section (VBAC) is safe and should be recommended for women with one previous CS, however, its practice in Bangladesh is low. VBAC has the potential to help reduce unnecessary CS in Bangladesh. As obstetricians play a significant role in birth decision-making, their opinions, and perspectives on barriers to VBAC need to be explored. This study will address a gap in the literature exploring barriers and enablers to promoting VBAC from the level of the obstetric decision-maker. This qualitative study was conducted in the Dhaka Division of Bangladesh in July 2023. Criterion sampling was used to select obstetricians for in-depth semi-structured interviews. Seven interviews were conducted in a private hospital in Dhaka city and five interviews were conducted in a non-governmental organisation (NGO) hospital outside Dhaka city. Ethical approval was received from the relevant organisations in both Liverpool and Bangladesh. The thematic analysis gave rise to three main themes: "policy awareness and national situation", "reasons for practice decisions" and "ways to improve service delivery". Despite good awareness of VBAC policies and appreciation of its benefits, obstetricians expressed a preference for repeat CS. From the perspective of obstetricians, the main barriers to VBAC practice are related to the structure and function of the health system. To create an environment that will enable safe practice of VBAC, health system improvement and community awareness of the benefits of normal vaginal birth are required.

the manuscript. Individual anonymised transcripts can be provided upon request.

**Funding:** The authors received no specific funding for this work.

**Competing interests:** The authors have declared that no competing interests exist.

## Introduction

The rate of caesarean section (CS) has been increasing globally over the last 30 years with 21% of all births estimated to be by CS [1]. The ideal rate of CS is much debated, and the focus of the current recommendations from the World Health Organization (WHO) is on optimising its use. They no longer set a recommended CS rate, however, population-level rates of CS greater than 15% are not associated with improvement in maternal or neonatal outcomes [2].

Access to CS both "too little too late" and "too much too soon" can hinder efforts to improve maternal and neonatal mortality rates [3]. The unmet need for Emergency Obstetric care (EmOC), including CS, remains a significant issue contributing to maternal mortality in many Low and Middle-Income countries (LMICs) [4]. On the other hand, an increase in facility-based births has contributed to over-medicalisation and unnecessary use of CS. In many countries, these two extremes of CS provision coexist resulting in a double burden of CS. While poorer members of the community still face challenges accessing EmOC, those who are wealthier are facing CS performed when not medically necessary [5].

One country that is facing this double burden of CS use is Bangladesh. The CS rate increased tenfold between 2000 and 2018 and currently 45% of births are by CS in Bangladesh according to the latest DHS survey [6]. The focus of Bangladesh's maternal health policy for years has been on increasing facility-based births to reduce preventable maternal deaths. The number of women giving birth in a health facility rose from 9.6% to 64.8% between 2000 and 2022, however this has been accompanied by a "caesarean epidemic" and has not resulted in the expected improvement in maternal outcomes [6].

Between 2000 and 2010, the maternal mortality ratio (MMR) dropped significantly from 322 to 194, as facility-based births and access to EmOC increased [7]. This decrease is also associated with other factors including improved access to contraception and reduced fertility rates. However, between 2010 and 2016 there was no improvement in MMR, raising concerns about the effectiveness of the health systems and quality of care [7].

The rate of CS in Bangladesh is significantly higher in private hospitals, where an average of 81.1% of births are by CS [8]. As the socio-economic status of women in Bangladesh improves, more women are giving birth in these private facilities. While women who are better educated, of higher socio-economic status and living in urban areas experience over-medicalisation of pregnancy and birth, other groups in the population are facing difficulties accessing appropriate care. In 2018, despite 300,000 women not being able to access CS when needed, 77% of all CS were performed without a medical indication [9]. This burden of unnecessary CS has both medical and economic consequences for mothers and their families and puts unnecessary demand on the already resource-limited health service [10].

As the rate of CS increases, more women face the decision between repeat CS and vaginal birth after caesarean (VBAC) for subsequent pregnancies. Based on current research, there is a consensus among international organisations that planned VBAC is safe for women with one previous CS with a low transverse scar [11]. According to the Bangladesh guidelines on VBAC, planned VBAC is successful in between 72–76% of cases [12]. Despite this, the "once a caesarean, always a caesarean" approach still appears to be common practice [13]. Begum et al. [14] found that previous CS was the most common indication for performing CS, reported in 35% of cases.

While both VBAC and repeat CS have their risks and benefits, the decision on the mode of birth should consider context-specific factors and involve the patient and family as well as obstetricians. In Bangladesh, research suggests that obstetricians have a particularly significant role in decision-making, with only 12% of women reporting making their own birth decisions

[15]. Therefore, to establish the specific barriers and enablers of VBAC practice in Bangladesh, the views of obstetricians need to be explored.

This study will address a gap in existing literature. Although the rate of CS in Bangladesh has received a lot of attention from researchers and policymakers, the available literature is mainly focused on primary CS and factors associated with higher rates. Of the studies that look at VBAC and repeat CS in Bangladesh, the focus is on the outcomes rather than the decision-making process, with a significant lack of qualitative research in the area. This study provides insight into the practice and experiences of the obstetric decision-maker working in the Dhaka Division. As the rate of CS continues to rise in Bangladesh, these findings could help inform policy and identify strategies to reduce the number of repeat CS performed.

### Aims and objectives

The overall aim of this study is to explore the views of obstetricians working in the Dhaka Division of Bangladesh on the barriers and enablers to promoting VBAC, with the following objectives:

- To explore clinicians' understanding and experiences of VBAC

- To identify strategies to help promote/ facilitate VBAC

## Methods

### Study design

This study was conducted using a qualitative approach, as the aim was to understand human behaviour and what influences it. This approach aligns with the interpretivist philosophy, which emphasises the understanding of individuals' subjective experiences [16]. Data was collected using face-to-face in-depth interviews, allowing participants the freedom to express their own opinions and experiences within a setting that is natural to them [17]. To analyse the data, thematic analysis as outlined by Braun and Clarke [18], was used due to its flexibility and ability to systematically identify, and present themes and patterns within qualitative data.

**Setting and participants.** The study was conducted in Bangladesh in July 2023. All data collection and analysis were conducted by the author, with help from a supervisor at Liverpool School of Tropical Medicine (LSTM) and a team at the Centre for Injury Prevention and Research Bangladesh (CIPRB).

Data collection took place in two different districts within the Dhaka Division. Seven interviews were conducted in a private hospital in Dhaka city and five interviews were conducted in a non-governmental organisation (NGO) hospital outside Dhaka city. The study hospitals were chosen with the help of senior staff at CIPRB as they have in-depth knowledge of the Bangladesh context and were helping facilitate this research.

**Sampling and recruitment.** Data collection continued until saturation was reached and no new themes were emerging. A total of 12 interviews were conducted. This is comparable to sample sizes in similar studies by Foureur et al. [19], and Jimenez-Zambrano et al. [20], implying the sample size was adequate and the study design was robust.

To account for potential variation in ideas and knowledge among Obstetricians with varying levels of experience, a varied sample was selected. A criterion sampling approach was used with the aim of interviewing at least one from each category (Table 1). This helped ensure no themes were missed and saturation had been reached. It also reduced the risk of selection bias, by avoiding the overrepresentation of one group, to produce a more comprehensive set of data. Due to the demanding schedules of obstetricians in Bangladesh, inclusion criteria were

**Table 1. Participant selection criteria.**

| Participant criterion | Minimum number of participants |
|---|---|
| Obstetricians<br>• Senior obstetrician<br>• Junior obstetrician<br>• obstetrician with experience working in another country/setting (where VBAC is practiced)<br>• working in the private sector<br>• working in the public sector<br>• someone involved in education and specialist training<br>• representative of the Obstetrical and Gynaecological Society of Bangladesh (OGSB) | 8 (at least 1 from each category) |
| Representative of the OGSB society | 1 |
| An individual involved in education and specialist training | 1 |
| **Total number** | 10–15 |

those available at the time of the interview, who fulfilled the criteria and agreed to be interviewed in English.

## Data collection

Data collection involved in-depth semi-structured interviews, conducted face-to-face by the author. All interviews took place within the hospital in a private room in the presence of the research assistant who was fluent in Bangla and English.

A topic guide was used as a starting point to ensure all the main ideas were discussed, however, the specific probes and prompts used varied between interviews. Interviews were audio recorded and transcribed verbatim by the interviewer within 48 hours. Notes were made by the research assistant during the interview, to avoid disrupting the flow of the interviewer.

A pilot interview was conducted in the UK prior to travel to ensure the questions were logical and the answers addressed the objectives. This was with an experienced obstetrician with clinical, teaching and research experience in LMICs working at LSTM. The topic guide was then discussed with the research team at CIPRB prior to starting interviews to ensure the questions were context appropriate. After data collection was completed, two Bangladeshi maternal health experts with extensive knowledge of the local context were also interviewed as key informants to better contextualise the findings. These experts work for highly reputable international organisations and were identified with guidance from senior staff at CIPRB, who have a deep understanding of the relevant issues.

## Analysis

All the transcripts were read several times for familiarisation and then coded with NVivo12 software to help organise the data. Analysis was iterative with codes generated both deductively based on the topic guide and inductively as new ideas emerged. Once all data was coded it was organised into themes. Initially, the themes were grouped based on the study objectives, however following review, and discussion this was adapted taking a more analytical approach.

## Quality assurance

To ensure trustworthiness, the principles of credibility, confirmability, dependability and transferability [21] were applied to this research. Table 2 outlines how potential threats to trustworthiness were mitigated.

**Table 2. Steps taken to mitigate threats to trustworthiness.**

| The potential threat to trustworthiness | How it was mitigated | Trustworthiness principle |
|---|---|---|
| Interviews conducted in English. | Ensuring participants were happy to be interviewed in English. Restricting participants to obstetricians who have received training in English. Presence of research assistant fluent in Bangla and English at all times. | Credibility, conformability |
| Social-desirability bias due to participants knowing each other. | All participants were aware participation was voluntary. Participants chose location of the interview to ensure they felt comfortable and not likely to be overheard. Data collection involved individual interviews rather than focus group discussions. | Credibility, conformability |
| Asking leading questions | Piloting and discussion of the topic guide, to allow time for adjustment before starting interviews. | confirmability |
| Personal influence on data collected | Being mindful of positionality. Keeping a reflective diary throughout the interview process. | confirmability |
| Misinterpretation of results | Discussion of the results with the other people present during interviews. | |

**Positionality.** The interviews were approached from a medical perspective, due to the researcher's background as a medical student. This was thought to enable a professional connection to develop during the interviews. However, the knowledge of the researcher on the technical aspects of VBAC and obstetrics was much lower than that of the interviewees. As a result, participants may have been less willing to disclose information, which they felt the interviewer may not understand. However, a lack of prior knowledge of the subject helps ensure an objective approach and reduces personal influence on the data collection. The lack of experience in other LMIC health systems and knowledge of the local language and culture, could also have influenced the positionality of the researcher. Support from senior staff at CIPRB and LSTM throughout helped mitigate this.

## Ethical considerations

**Confidentiality and informed consent.** Once transcribed, data was pseudo-anonymised to protect the confidentiality of participants and all data was stored on a password-protected computer and saved to LSTM OneDrive. All participants were given the opportunity to read the consent form and participant information sheet. Participants were informed that data would be kept anonymous, participation was voluntary, and they could withdraw at any time until transcription. All the forms were in English. This was not anticipated to be a problem as participants had all received their medical training in English, however, the research assistant was present during the consent and interview process to provide clarifications in Bangla if necessary.

**Sensitive issues.** In accordance with internationally accepted principles of research ethics, the overall aim was to ensure no harm to participants. As the study involved exploring decision-making and past experiences, there was potential for sensitive conversations which may have caused distress or feelings of judgment. To mitigate this, questions on the topic guide did not involve individual practice and participants were asked not to discuss specific cases.

**Ethical approval.** Ethical approval for this study was obtained from both the Liverpool School of Tropical Medicine MSc Ethical Review Panel; and the Centre for Injury Prevention and Research Bangladesh (CIPRB) Ethical Review Committee.

## Results

Table 3 shows the participant characteristics. Most participants were female, reflecting the demographics of the obstetric workforce in Bangladesh. The majority were consultants, as trainees were much harder to locate due to being busier with patients. Details about experience

**Table 3. Participant characteristics.**

| Participant characteristics | | Number of participants |
|---|---|---|
| Hospital site | Dhaka City (Private) | 7 |
| | Outside Dhaka City (NGO) | 5 |
| Level of training | Consultant | 10 |
| | Registrar | 2 |
| Sex | Male | 2 |
| | Female | 10 |
| Experience in: | Other country | 3 |
| | Public sector post-graduation | 5 |
| | Medical education | 8 |
| | Research | 4 |

in the public sector and other countries were also collected as well as previous work in education or research.

Three main categories were identified from the analysis of the results: "policy awareness and national situation", "reasons for practice decisions" and "ways to improve service delivery" with various themes in each category (Fig 1).

## Policy awareness and national situation

**Current VBAC situation.** Overall, the perspectives of VBAC were positive. VBAC can reduce the cost and length of hospital stay and prevent complications associated with multiple

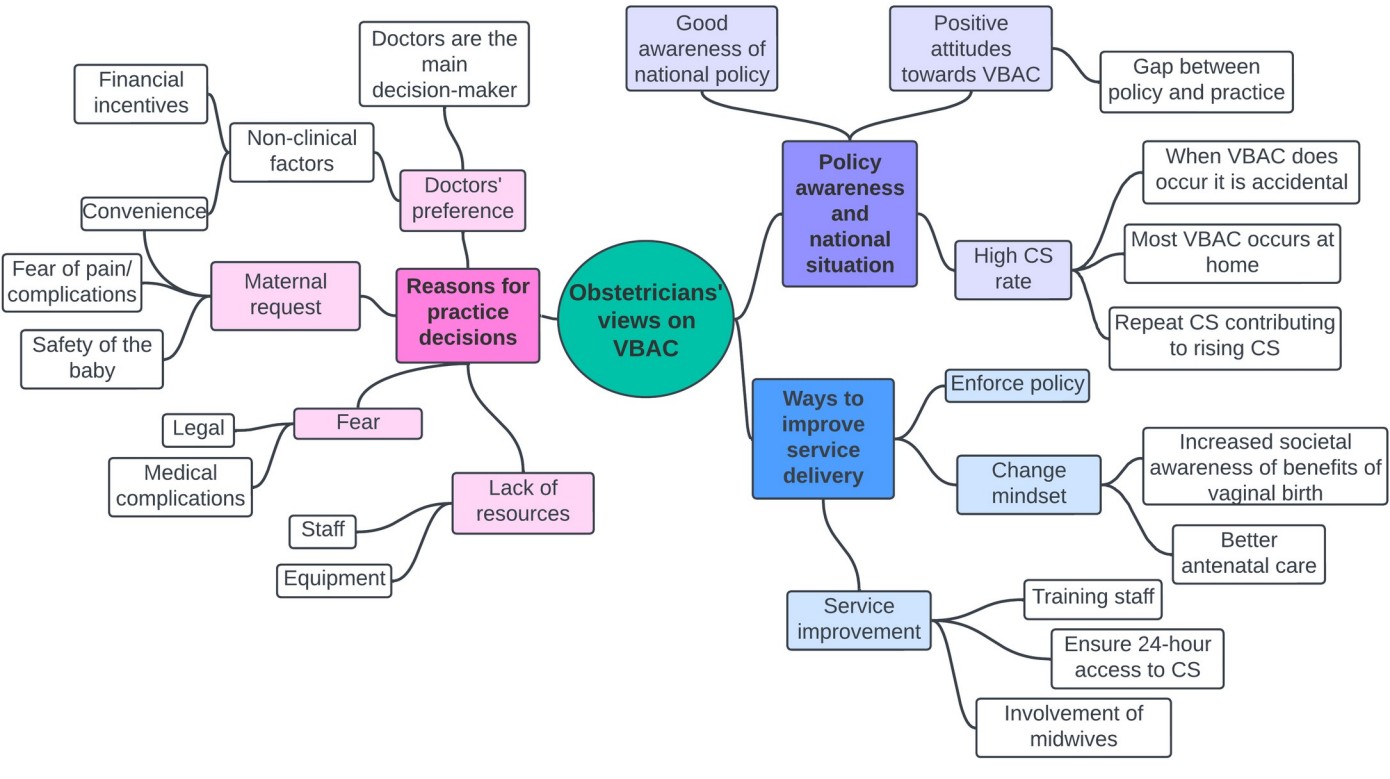

**Fig 1. Concept map showing themes identified.**

CS. The risk of prematurity, placenta accreta, risk of haemorrhage and chronic pain caused by adhesions were all mentioned by participants as negative effects of multiple CS. The consensus among most participants across both study sites was that the level of practice of VBAC is lower than it should be.

Although most participants were aware of the national VBAC policy, they reported a mismatch between policy and practice. It was clear from both hospitals that VBAC is not routinely planned for, nor practised. When it does occasionally occur, it is often accidental. Many participants described similar past experiences where a woman with a previous CS presented to the hospital in established labour.

*"by chance they deliver normally. . . patient lies down and delivers. After that she says oh I have got one caesarean section" (Registrar, NGO hospital)*

The lack of practice of VBAC in hospitals was also partly attributed to the high rate of home births in the country. Obstetricians stated that many women, including those with a previous CS, choose to give birth vaginally at home. As a result, women who attend hospital are often those who have experienced complications and are therefore more likely to require a CS. Participants also expressed a lack of confidence in performing assisted vaginal delivery (AVD), due to insufficient training and negative past experiences. As a result, CS is often performed as the first alternative to vaginal birth should a complication arise.

**Repeat caesarean section.**   The estimated proportion of CS performed with "previous CS" as an indication varied from 50 to almost 100%. When asked how many women with a previous CS undergo VBAC the answer was unanimously very few. While participants acknowledged that an increase in VBAC would be beneficial in reducing unnecessary CS, the priority of many clinicians was to address high rates of primary CS. Fewer primary CS would in turn reduce the need for repeat CS.

## Reasons for practice decisions

*"The outcome will be fantastic, but I cannot make is safe so don't do it" (Consultant, NGO Hospital)*

**Fear.**   The main barrier to VBAC practice, present throughout all the interviews, was fear. This manifested in different ways from concerns about the safety of the baby to the fear of litigation should there be any adverse outcomes.

The most frequently identified complication of VBAC was the risk of uterine rupture. Due to the unpredictable nature of the potential complications of VBAC, many obstetricians preferred elective CS as it created a sense of control and was associated with less "risk".

*"though the risk [of rupture] is not so high, the perception of obstetricians is that VBAC is not safe" (Consultant, NGO Hospital)*

As well as fear of medical complications, clinicians also expressed concerns that mothers or their families may blame doctors for not intervening should there be an adverse outcome from attempted VBAC. Clinicians felt that by performing a CS, the family are more likely to accept that the obstetricians have done everything possible.

*"if there are any complications then they will think it is wrong, actually no it is a complication of normal delivery. Misunderstanding, miscommunication. So to avoid risk, we do caesarean section." (Consultant, Private Hospital)*

As well as the response from families, the lack of legal protection and thorough investigation of adverse outcomes was also stated as a barrier to performing VBAC. Obstetricians expressed fear of litigation and the potential impact this may have on the reputation of both themselves and the hospital.

*"doctors are going to the jail before any investigation, so that is very much tragic" (Consultant, Private Hospital)*

**Lack of resources.** Many reasons behind the fear of complications and perceptions of increased risks stemmed from a lack of resources. Lack of staff and equipment were barriers to making VBAC safe and being able to manage any complications should they arise.

To practice VBAC according to the guidelines, close monitoring of the mother and fetal heart rate is required. While not specifically required in the Bangladesh VBAC policy, without access to cardiotocography (CTG), more than half of the clinicians did not feel comfortable that they could timely identify when emergency CS was necessary. Lack of trained staff to monitor patients was also a concern.

*"doctor-patient ratio is more and it is a highly populated country and we are overloaded so we have less time for the patient to monitor" (Consultant, Private Hospital)*

As well as close monitoring, emergency CS should be available immediately if needed. Despite being a requirement for facilities providing comprehensive EmOC (CEmOC) services, several obstetricians from both hospitals expressed concerns that emergency CS was not available 24 hours.

*"If any problem, then we take care of the patient and CS done quickly. This is not possible in our hospital." (Consultant, Private Hospital)*

The lack of consultant support at night, availability of anaesthetists and competition for operating theatres all caused concern about timely access to emergency CS when needed. In both hospitals, consultants are present from 8 am until around 3 pm, after this time a consultant will be on call but not necessarily present in the hospital. Should an emergency arise after this time, it is often medical officers who oversee decision-making.

Obstetricians expressed concern that medical officers' lack of experience can cause delays in decision-making as well as substandard care if they are required to perform an emergency CS. Emergency CS is more complicated than elective CS and when performed by someone with less experience, the risk of complications could be higher. Obstetricians reported a preference for scheduling elective CS during office hours to prevent this risk.

*"Who will do the CS at this time, the medical officer. Now medical officer will do a complicated section when head is down, it is not an easy operation" (Consultant, NGO Hospital)*

As is the case with obstetric consultants, anaesthetists were not present in the hospitals at night. This can be hugely problematic as the time taken for them to travel to the hospital can cause serious delays and was a barrier to enabling safe VBAC.

*"I call the anaesthetist, he will come from the other side of the river, the boat is not there so half an hour he is waiting to cross the river. Then he will come to OT, in the meantime rupture happen, baby die"* (Consultant, NGO Hospital)

**Doctors' preference.** Doctors, particularly consultants, were reportedly the main decision-makers regarding the mode of birth. When asked who is involved in the decision-making process, the patient or patient's family was only mentioned by two participants without prompting.

As the process of vaginal birth is unpredictable and can take a long time, several clinicians expressed a preference for CS as it is more convenient. Elective CS not only gives doctors more time to see other patients but also allows planning to ensure all the necessary resources are available.

*"Elective anything is positive. Why? I can ask my friend, come here I may need blood, in the early morning having a cup of tea, the surgeon will do the procedure in a high mood, blood bank is in a high mood, patient is still having a fresh bath with a good dress clean dress. Obviously, this is the positive"* (Consultant, NGO Hospital)

Another factor contributing to the preference for elective CS is financial incentives. While this was identified as a barrier by two participants from the NGO Hospital, it was not directly mentioned by participants working at the Private hospital.

*"This is a corporation, not a service. It is a business. Health sector should not be a business"* (Consultant, NGO Hospital)

The higher rate of CS in the private sector was identified by both consultants and registrars; however, more junior staff were reluctant to elaborate on this and details were only given by more senior consultants. This is likely due to the hierarchal nature of the health workforce, with more junior staff fearful of repercussions.

*"CS is higher in private sector. . . The exact reason why I cannot mention"* (Registrar, NGO Hospital)

**Maternal request.** Another barrier to VBAC reported by obstetricians was maternal request for caesarean section to avoid the perceived risk of attempting VBAC.

*"We counsel the patients about the risk factor of the patient for VBAC they deny [decline]. This is the most common barrier to conduct the VBAC here"* (Consultant, Private Hospital)

Mothers also reportedly request CS for convenience and fear of vaginal birth. According to obstetricians, mothers prefer elective CS as it avoids hours of labour pain and complications of vaginal birth such as episiotomy. To many, CS is seen as a quick procedure for which they have anaesthetic and therefore don't feel any immediate pain.

*"people nowadays are loving the caesarean section. The educated people, they think it is better to do a caesarean section for half an hour than waiting for labour." (Consultant, NGO Hospital)*

## Ways to improve service delivery

**Mindset.** The most popular strategy to help increase the practice of VBAC was to change the mindset of mothers, both through improved antenatal counselling and raising awareness of the benefits of normal vaginal birth. Many women in Bangladesh currently do not receive proper antenatal care and therefore may only see a medical professional during birth. One participant suggested counselling women about birth options for subsequent pregnancies at the time of the first CS. This way, women are aware of the birth options at the start of the next pregnancy irrespective of whether they receive antenatal care.

**Health system improvement.** Several participants expressed concerns that lack of training can cause delays in identifying complications and performing emergency CS when necessary, leading to a lack of trust between consultants and more junior members of staff.

*"Sometimes new doctors come, and they cannot recognise the patient's ruptured uterus when occurring . . .then the problems can arise" (Consultant, NGO Hospital)*

An increase in manpower was also suggested to promote VBAC. One strategy identified was to increase the number of midwives and better integrate them into the provision of intrapartum care. By allowing trained midwives to manage uncomplicated normal vaginal births, clinicians stated they would then have more time to manage other patients such as those attempting VBAC.

**Enforce policy.** While most obstetricians were aware of the existence of the national VBAC guidelines, they were largely not being implemented. Providing training for doctors on these guidelines would encourage a common approach to practice and strengthen confidence in VBAC. The guidelines however are largely based on international recommendations and do not consider the clinical context. Several clinicians identified the need for improvement in health infrastructure, such as the availability of 24-hour emergency CS before these guidelines can be safely implemented.

## Discussion

The results of this study have highlighted the need for health system improvement to create an enabling environment for safe VBAC. To systematically analyse the implications of the study findings, the WHO health systems "building blocks" framework will be used (Fig 2).

### Service delivery

Many women in Bangladesh still give birth at home and may only attend a healthcare facility if there are complications. This was identified as the reason for the high institutional CS rate in Bangladesh. While this may be reasonable to assume, data suggests that most CS performed are elective, especially in private facilities where 47% of births are by elective CS [8]. Maternal request was identified by many obstetricians as a significant factor influencing their practice decisions. In many cases, women may attend private facilities specifically for a CS. Similar findings were found by Mia et al. [23], who identified maternal requests as the main reason for increasing CS rates in private facilities. As this study focused on the views of obstetricians and not the mothers themselves, it is difficult to establish how valid these claims are and whether it

**Service delivery**

**Health workforce**

**Health information systems**

**Access to essential medicines**

**Financing**

**Leadership/ governance**

**Fig 2. The WHO health system building blocks (from [22]).**

is representative of all women. This also raises questions over how women are counselled on birth options after a previous CS and whether VBAC is discussed as an option. As supported by previous literature by Mainuddin et al. [15] and Sathyanarayanan et al. [24], obstetricians are the main decision-makers in Bangladesh, which is important to consider when evaluating these claims. The fact obstetricians reported factors such as convenience and financial

incentives as reasons for CS, raises questions over motives when counselling women about birth options after a CS.

Obstetric complications are unpredictable, so by its definition, a CEmOC facility should be able to provide CS at any time whenever a woman needs it. The lack of anaesthetists and obstetric consultants 24/7 was a significant barrier to being able to provide 24-hour emergency CS, thereby preventing obstetricians from feeling confident to safely practice VBAC. This is supported by the results of the Bangladesh Health Facility Survey which found that only 44.7% of health facilities in the country providing intra-partum care had trained staff present at all times [25]. Lack of access to EmOC services 24/7 will have profound impacts on maternal and neonatal outcomes and is likely contributing to the stall in progress in reducing maternal mortality in Bangladesh. Previous studies have suggested that private facilities have better readiness for EMOC than public facilities [26], however, the results of this study indicate that poor availability of EmOC services is a health system-wide problem.

## Health workforce

Despite receiving VBAC training from the Obstetrical and Gynaecological Society of Bangladesh (OGSB), obstetricians did not feel confident that everyone in the team had the skills to be able to monitor a woman undergoing a VBAC and identify signs of complications. Trust among healthcare providers is an integral part of being able to provide high-quality care. As well as fear of medical complications, fear of legal repercussions also dictates practice decisions. There appeared to be a lack of understanding by the public that birth complications are often inevitable and not necessarily due to negligent care. Due to this apparent "blame culture", doctors are hesitant to accept the risks associated with VBAC due to fear of damaging their professional reputation and that of the hospital. This finding supports the evidence from other VBAC studies in Iran and the Netherlands where medico-legal concerns were identified as barriers to VBAC [27, 28].

Although fear of litigation is clearly not a problem isolated to LMICs, the consequences of legal action can be much more impactful when doctors lack the support systems present in many high-income countries. These issues were particularly prominent in the context of recent events that had taken place in Dhaka. At the time of the interviews, there was a series of protests in response to the arrest of two obstetricians following the death of a mother and baby. The arrests were made following claims of medical malpractice by the patient's family without any internal investigation. It is therefore understandable why obstetricians were wary of taking any sort of risk.

As well as the lack of VBAC training, a lack of skills to be able to safely perform assisted vaginal delivery (AVD) was also identified. Evidence suggests that while in High-Income countries the rising CS rate is accompanied by a similar increase in AVD, this is often not the case in LMICs [29, 30]. Lack of skilled operators, functioning equipment and fear of complications and litigation have all been shown to prevent AVD in LMICs. AVD can help reduce unnecessary CS and is associated with fewer complications, faster recovery, and shorter hospital stays. Outcomes that are particularly important in settings like Bangladesh where complications of CS are more common and out-of-pocket expenses are the greatest contributor to health spending [31, 32]. The lack of AVD is relevant for the low VBAC practice, but also likely contributes to the number of primary CS.

## Health information systems

The results of this study support the existing literature that repeat CS is a significant factor contributing to Bangladesh's rising CS rates. The huge variation in obstetricians' estimates of the

number of CS performed with "previous CS" as an indication suggests a lack of robust data reporting for CS. This issue was also identified by Begum et al. [33], who found that for 22% of CS performed, there was no recorded indication. Proper recording of CS indication is essential to be able to identify women who had a primary CS for a non-recurring reason and therefore may be eligible for VBAC in subsequent pregnancies.

## Financing

Non-clinical factors including convenience and financial incentives also influence practice decisions, particularly in the private sector. These factors were more readily discussed by participants working in the NGO hospital, likely because it does not function on the same for-profit basis as private facilities. Poor regulation of private sector CS costing has created a business-like model of service delivery. Based on a key informant interview with a sexual and reproductive health systems expert, the costs for CS vary between 1,500 and 30,000 Taka (around £10 to £220) whereas vaginal birth usually costs less than 12,000 Taka (£90). Other studies indicate these costs may be even higher. Khan et al. [34] found an average cost of around £480. In a study in Iran, despite the payment being higher for VBAC than CS, doctors still did not feel the extra money was worth the longer time required to monitor a woman during normal vaginal birth [27]. The influence of these factors on clinical decision-making risks breaching the principles of medical ethics, as obstetricians may not be acting in the patient's best interests.

## Leadership/ governance

This study showed that obstetricians' awareness of national policies and guidelines was good; however, their effective implementation remains challenging. Healthcare in Bangladesh is provided by four key actors: government, private, NGO and donor agencies. The fragmented nature of the health system makes it difficult to establish responsibility and accountability. Lack of regulation in the private sector is a major concern, identified by the government maternal health policy as well as in other national studies [35, 36]. Lack of regulation has resulted in significant variation in the quality of care provided, compromising improvements in maternal outcomes. NGO-run healthcare facilities have also been shown to lack accountability to the government [37], demonstrating the need for effective national health system regulation.

The centralisation of the health system is also problematic. Ultimate authority lies with the Ministry of Health and Family Welfare which results in a disconnect between centralised policymaking and the needs of the population. This was apparent from the results of this study which found that current policies do not consider the context and therefore their implementation is not currently feasible. This top-down approach can also make healthcare provision susceptible to political influence, as explored by Islam [38]. The motives and priorities of the decision-makers may not align with the needs of the population. Decentralisation of policymaking and better health system monitoring will help promote the standardisation of healthcare provision.

## Limitations

As all the interviews were conducted in the Dhaka division, the results are not necessarily transferable to other regions of the country. However, as many of the barriers to VBAC stemmed from health system issues, these findings may also be relevant elsewhere in the country. Further research is needed to determine the wider applicability of these results. Although many participants had experience working in government hospitals, these findings may also

not represent the situation in the public sector. However, as the rate of CS is significantly higher in private facilities, the choice of study sites was felt to be appropriate.

## Recommendations

Evidence shows that engaging with communities is crucial to improving healthcare access and provision to ensure universal health coverage [39]. Therefore, these recommendations address both the community and the health system building blocks (Fig 2).

### Community

Increased awareness of the effects of multiple CS and the benefits of vaginal birth is vital. As many women in Bangladesh lack autonomy over healthcare decisions, a community-wide approach is necessary. Historically, health education programmes have not accounted for the low literacy levels in the country, and have therefore only reached the educated section of the population [40]. Ensuring health education is accessible to individuals regardless of their level of education is vital to reduce health disparities and empower people to make informed healthcare decisions.

### Health service delivery

Without a supportive and safe environment, practice of VBAC will likely remain low. Ensuring 24-hour availability of CEmOC services including CS should be a priority on the maternal health agenda. The poor function of public CEmOC facilities is identified by the government's national strategy for maternal health [36], however as more women choose to give birth in private facilities, the focus needs to be widened to ensure these facilities are also functioning to an acceptable standard.

### Health workforce

Training of staff not only on the technical aspects of VBAC but also on decision-making for CS and would encourage a common approach to practice and improve patient outcomes. Policymakers should also consider it a priority to improve skills and practice of AVD, to help enable VBAC and address rising CS rates.

As the new cadre of midwives is rolled out in Bangladesh, there needs to be a multi-disciplinary approach to maternity care. When well-trained and working in a supportive and enabling environment, midwives can manage uncomplicated vaginal births and support women with VBAC, allowing obstetricians more time to provide specialist care as needed. As with any new profession, there is a risk that other healthcare staff and society will lack trust in their skills. Therefore, it is also important to raise awareness of the role and benefits of midwives.

### Health information systems

The current 'blanket approach' of repeat CS to manage all women with previous CS highlights the need for improved reporting of the reasons the first CS was performed. If women who had a primary CS for a non-recurring reason can be easily identified, they can be provided with proper counselling on the benefits of VBAC.

The need for improvement in data recording for other indicators is also a priority. Ensuring accurate data collection is important to monitor improvement and ensure interventions lead to the desired result. Despite the rising CS rates not being associated with improvements in maternal mortality, data on the number of deaths because of CS is not available. To be able to

monitor improvements in VBAC practice, the number of eligible women who undergo a planned trial of VBAC also needs to be recorded.

### Leadership/ governance

There is currently a lack of adherence to the national VBAC policy. Strengthening policy to ensure recommendations are context-specific is necessary to ensure implementation is feasible.

To promote a more supportive environment for obstetricians and build a sense of trust among colleagues, a system of mentoring and accountability is also important. This will ensure staff have confidence in each other's skills and are able to learn from poor practice decisions. Advocating for better legal protection for doctors will also help reduce the fear of complications among doctors and improve doctor-patient relationships.

## Conclusion

This study addresses the lack of qualitative data on VBAC in Bangladesh by helping understand the barriers to practice from the level of the decision-makers. The results of this study can help inform policy and service improvement to enable the safe practice of VBAC in Bangladesh as a strategy to reduce unnecessary CS.

The results show that planned VBAC is rarely practised in the Dhaka Division. As medical professionals, obstetricians' decisions should be evidence-based, however, this needs to be contextualised. The current health system is not capable of creating an environment that facilitates the safe practice of VBAC, hence CS appears the safest option. Non-clinical factors such as maternal preference, convenience and financial incentives are also contributing to a general preference for CS over vaginal birth.

Promoting community engagement and improving the healthcare system to ensure 24-hour access to EmOC is vital to creating an enabling environment for VBAC. Increased practice of VBAC will help reduce unnecessary CS and improve maternal health outcomes as a result.

## Supporting information

**S1 File. Topic guide for in-depth interviews.**
(DOCX)

**S1 Checklist. Inclusivity in global research.**
(DOCX)

## Acknowledgments

We would like to thank everyone who participated in this study. We are also grateful to the staff at OGSB, UNFPA and CIPRB for their guidance and expertise.

## Author Contributions

**Conceptualization:** Aimee Hairon, Terry Kana.

**Data curation:** Aimee Hairon, Sumaiya Afroze Khan Atina.

**Formal analysis:** Aimee Hairon.

**Investigation:** Aimee Hairon, Abu Sayeed Md. Abdullah, Sumaiya Afroze Khan Atina, Terry Kana.

**Methodology:** Aimee Hairon, Abdul Halim, Abu Sayeed Md. Abdullah, Terry Kana.

**Project administration:** Aimee Hairon, Abdul Halim, Abu Sayeed Md. Abdullah, Sumaiya Afroze Khan Atina, Terry Kana.

**Resources:** Abu Sayeed Md. Abdullah, Terry Kana.

**Software:** Aimee Hairon.

**Supervision:** Abdul Halim, Terry Kana.

**Validation:** Abdul Halim.

**Writing – original draft:** Aimee Hairon.

**Writing – review & editing:** Aimee Hairon, Abdul Halim, Terry Kana.

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
