## [Decision Letter · Decision Letter 0]

2 Jul 2024

PGPH-D-24-00771

"I can't make it safe, so I don't do it": Exploring Obstetricians' views on barriers and enablers to promoting vaginal birth after Caesarean Section in Bangladesh

Dear Dr. Hairon,

Thank you for submitting your manuscript to PLOS Global Public Health. After careful consideration, we feel that it has merit but does not fully meet PLOS Global Public Health’s publication criteria as it currently stands. Therefore, we invite you to submit a revised version of the manuscript that addresses the points raised during the review process.

Please note that we have only been able to secure a single reviewer to assess your manuscript. We are issuing a decision on your manuscript at this point to prevent further delays in the evaluation of your manuscript. Please be aware that the editor who handles your revised manuscript might find it necessary to invite additional reviewers to assess this work once the revised manuscript is submitted. However, we will aim to proceed on the basis of this single review if possible. 

The reviewer has requested clarification regarding several aspects of the methodology. Could you please revise the manuscript to carefully address the concerns raised?

Please also provide a copy of your interview guide as a "supporting information" file.

We look forward to receiving your revised manuscript.

Kind regards,

Steve Zimmerman, PhD

PLOS Staff Editor

Journal Requirements:

2. Please include a complete copy of PLOS’ questionnaire on inclusivity in global research in your revised manuscript. Our policy for research in this area aims to improve transparency in the reporting of research performed outside of researchers’ own country or community. The policy applies to researchers who have travelled to a different country to conduct research, research with Indigenous populations or their lands, and research on cultural artefacts. The questionnaire can also be requested at the journal’s discretion for any other submissions, even if these conditions are not met. Please find more information on the policy and a link to download a blank copy of the questionnaire here: https://journals.plos.org/globalpublichealth/s/best-practices-in-research-reporting. Please upload a completed version of your questionnaire as Supporting Information when you resubmit your manuscript.

3. In the online submission form, you indicated that "Excerpts of the transcripts relevant to the study are presented in the manuscript. Individual anonymised transcripts can be provided upon request". 

3. Uploaded as supplementary information.

Additional Editor Comments (if provided):

Reviewers' comments:

Reviewer's Responses to Questions

**Comments to the Author**

1. Does this manuscript meet PLOS Global Public Health’s publication criteria? Is the manuscript technically sound, and do the data support the conclusions? The manuscript must describe methodologically and ethically rigorous research with conclusions that are appropriately drawn based on the data presented.

Reviewer #1: Partly

2. Has the statistical analysis been performed appropriately and rigorously?

Reviewer #1: N/A

3. Have the authors made all data underlying the findings in their manuscript fully available (please refer to the Data Availability Statement at the start of the manuscript PDF file)?

Reviewer #1: No

4. Is the manuscript presented in an intelligible fashion and written in standard English?

Reviewer #1: Yes

5. Review Comments to the Author

Reviewer #1: The objectives are aiming to generalize the findings to Bangladesh but the methodology does not allow such a generalization. Revision is recommended.

The methodology section only states that a qualitative methodology will be used. However, the qualitative methodology is complex and a better description of what exact method and the philosophy behind the chosen methodology is required.

it appears that the researchers had decided on the sample size before the commencement of the study which is not a practice in qualitative research. However, reaching the point of saturation too is stated. Recommend editing the text appropriately.

Under participant selection, the authors have selected participants from different categories. However, it is not clear if there was saturation for each of these categories. Different categories may have different ideas and knowledge which may not be reflected in the interviews with other categories.

At the end of the data collection section, authors refer to experts in the subject who interviewed for triangulation purposes. However, no information on why they are considered as experts and how they were selected is provided.

The authors conclusions are for the whole of Bangladesh, however qualitative studies do not allow such generalizations.

The findings of the study are important and requires attention of policy makers and program managers.

6. PLOS authors have the option to publish the peer review history of their article (what does this mean?). If published, this will include your full peer review and any attached files.

**Do you want your identity to be public for this peer review?** For information about this choice, including consent withdrawal, please see our Privacy Policy.

Reviewer #1: No

---

## [Decision Letter · Decision Letter 1]

20 Oct 2024

PGPH-D-24-00771R1

"I can't make it safe, so I don't do it": Exploring Obstetricians' views on barriers and enablers to promoting vaginal birth after Caesarean Section in Bangladesh

Dear Dr. Hairon,

Thank you for submitting your manuscript to PLOS Global Public Health. After careful consideration, we feel that it has merit but does not fully meet PLOS Global Public Health’s publication criteria as it currently stands. Therefore, we invite you to submit a revised version of the manuscript that addresses the points raised during the review process.

The revised manuscript has been assessed by 2 reviewers. Overall they are content with the revisions however Reviewer 2 has recommended a minor revision for the results should be separated into sections by theme.  Please address this recommendation and resubmit a revised manuscript.  

We look forward to receiving your revised manuscript.

Kind regards,

Emma Campbell, Ph.D

Staff Editor

Journal Requirements:

Reviewers' comments:

Reviewer's Responses to Questions

**Comments to the Author**

1. If the authors have adequately addressed your comments raised in a previous round of review and you feel that this manuscript is now acceptable for publication, you may indicate that here to bypass the “Comments to the Author” section, enter your conflict of interest statement in the “Confidential to Editor” section, and submit your "Accept" recommendation.

Reviewer #1: All comments have been addressed

Reviewer #2: All comments have been addressed

2. Does this manuscript meet PLOS Global Public Health’s publication criteria? Is the manuscript technically sound, and do the data support the conclusions? The manuscript must describe methodologically and ethically rigorous research with conclusions that are appropriately drawn based on the data presented.

Reviewer #1: (No Response)

Reviewer #2: Yes

3. Has the statistical analysis been performed appropriately and rigorously?

Reviewer #1: (No Response)

Reviewer #2: Yes

4. Have the authors made all data underlying the findings in their manuscript fully available (please refer to the Data Availability Statement at the start of the manuscript PDF file)?

Reviewer #1: (No Response)

Reviewer #2: Yes

5. Is the manuscript presented in an intelligible fashion and written in standard English?

Reviewer #1: (No Response)

Reviewer #2: Yes

6. Review Comments to the Author

Reviewer #1: (No Response)

Reviewer #2: Great revisions, thank you! One minor comment might be to add sections to your result section by theme. Extremely minor but it might help to break up with results section.

7. PLOS authors have the option to publish the peer review history of their article (what does this mean?). If published, this will include your full peer review and any attached files.

**Do you want your identity to be public for this peer review?** For information about this choice, including consent withdrawal, please see our Privacy Policy.

Reviewer #1: No

Reviewer #2: **Yes: **Andrea Jimenez-Zambrano

---

## [Editor Report · Decision Letter 2]

31 Oct 2024

"I can't make it safe, so I don't do it": Exploring Obstetricians' views on barriers and enablers to promoting vaginal birth after Caesarean Section in Bangladesh

PGPH-D-24-00771R2

Dear Miss Hairon,

We are pleased to inform you that your manuscript '"I can't make it safe, so I don't do it": Exploring Obstetricians' views on barriers and enablers to promoting vaginal birth after Caesarean Section in Bangladesh' has been provisionally accepted for publication in PLOS Global Public Health.

Best regards,

Julia Robinson

Executive Editor